# Longer Participation in the Special Supplemental Nutrition Program for Women, Infants, and Children Is Not Associated with Reduced Sugar-Sweetened Beverage Intake among Black Participants

**DOI:** 10.3390/nu14051048

**Published:** 2022-02-28

**Authors:** Christopher E. Anderson, Catherine E. Martinez, Keelia O’Malley, Lorrene D. Ritchie, Shannon E. Whaley

**Affiliations:** 1Division of Research and Evaluation, Public Health Foundation Enterprises (PHFE) WIC, Irwindale, CA 91706, USA; catherinem@phfewic.org (C.E.M.); shannon@phfewic.org (S.E.W.); 2Department of Social, Behavioral and Population Sciences, School of Public Health and Tropical Medicine, Tulane University, New Orleans, LA 70112, USA; komalley@tulane.edu; 3Nutrition Policy Institute, Division of Agriculture and Natural Resources, University of California, Oakland, CA 94607, USA; lritchie@ucanr.edu

**Keywords:** WIC, sugar-sweetened beverage, race/ethnicity, disparities, water

## Abstract

This study assessed relationships of duration of family Special Supplemental Nutrition Program for Women, Infants and Children (WIC) participation with racial/ethnic disparities in child sugar-sweetened beverage (SSB) and water intake. Child beverage intake and family duration on WIC were collected during three cross-sectional surveys of WIC-participating families in Los Angeles County, California (2014, 2017 and 2020; *n* = 11,482). The associations of family duration of WIC participation, a proxy for the amount of WIC services received, with daily intake of total SSBs, fruit-flavored SSBs and water were assessed in race/ethnicity strata with multivariable negative binomial and Poisson regression models. Among English-speaking Hispanic children, those of families reporting 10 years of WIC participation consumed 33% and 27% fewer servings of total and fruit-flavored SSBs compared to those of families reporting 1 year on WIC. Among Black children, those from families reporting 5 and 10 years of participation in WIC consumed 33% and 45% more daily servings of fruit-flavored SSBs than those from families reporting 1 year on WIC. Disparities in daily total and fruit-flavored SSB intake between Black and White children increased with longer family duration on WIC. Duration of family WIC participation is associated with healthier beverage choices for infants and children, but does not appear to be equally beneficial across racial/ethnic groups in Los Angeles County.

## 1. Introduction

The prevalence of childhood obesity in the United States is high [1,2]. Recent data have suggested childhood obesity rates overall may have plateaued [3], but severe obesity continued to increase from 2013–2014 to 2016, with more pronounced increases for Hispanic girls and African American boys [4]. Preliminary data collected during the COVID-19 pandemic suggest that the rate of weight gain among children accelerated rapidly, adding further urgency to primary prevention efforts [5]. Childhood obesity prevalence is also not uniformly distributed across the population by socioeconomic status (SES); it is higher among children from low-income households [6], the majority of whom participate in the Special Supplemental Nutrition Program for Women, Infants and Children (WIC) at some point before they are 5 years of age [7]. WIC is a federal nutrition assistance program of the United States Department of Agriculture (USDA), serving pregnant and post-partum women, infants and children under 5 years of age in households with incomes below 185% of the federal poverty level (FPL). WIC provides supplemental healthy foods, health and social service referrals, breastfeeding support, nutrition education and individualized counseling. WIC is unique among federal nutrition assistance programs in the requirement that it provides nutrition education to participants, in addition to nutritious foods.

Children from low SES households are more likely to consume sugar-sweetened beverages (SSB) compared to children from high SES households, and Black and Hispanic children are >40% more likely to consume SSBs than White children [8]. Among children 12–24 months of age, 31.2% consume SSBs and 68.0% consume water on a given day, averaging 296 and 308 g/day among those who consumed each [8]. Most children 4 to 8 years of age consume less water than recommended [9], and Black and Hispanic families are more likely than White families to give children bottled water because of concerns regarding unsafe tap water [10], though racial/ethnic disparities in the amount of water consumed are not evident [8]. An analysis of the California Health Interview Survey indicates stable SSB consumption and persistent racial/ethnic disparities between 2013 and 2016 [11]. The proportion of children consuming any SSB on a given day increased from 23% among children aged 2–5 years to 37% among children aged 6–11 years [11]. Given these differences in early-life beverage choices, and the wide reach of the WIC program, WIC participation could contribute to efforts to address disparities by race, ethnicity and SES in early-life dietary patterns and obesity [12].

Associations of WIC participation and improved diet quality among children have been reported previously, particularly for consumption of the healthy foods provided as part of WIC supplemental food packages, such as fruits and vegetables [13]. WIC nutrition education, comprised of live classes, online classes, and individual counseling, includes information discouraging SSBs and encouraging water intake. Data have suggested that WIC-participant SSB consumption has decreased [14], and a recent study identified significant reductions in SSB intake among WIC-participating children of families with longer duration on the program [15]. The present study was conducted to assess whether SSB intake is reduced and whether water intake is increased with longer family duration on WIC *within* all racial/ethnic groups, and whether disparities in SSB and water intake *between* race/ethnicity groups would be reduced by longer family duration on WIC. It was hypothesized that children of all race/ethnicity groups would demonstrate lower daily SSB intake and higher daily water intake with longer family duration on WIC, and that racial/ethnic disparities in SSB and water intake would be reduced by longer family duration on WIC.

## 2. Materials and Methods

### 2.1. Setting and Sample

The sample for this study includes respondents to the cross-sectional Los Angeles (LA) County WIC Survey, a triennial phone survey conducted in a random sample of WIC participants from the largest local population of WIC participants in the nation. A county-wide random sample was augmented with random samples within 14 specific communities (2014, 2017 and 2020) and Asian and Black families across the county (2020). The survey aids planning efforts by WIC and community organizations serving LA County families with children under 5 years of age. This study includes data from 11,482 WIC participants 4 to 59 months of age, collected during the 2014 (*n* = 3644), 2017 (*n* = 3658) and 2020 (*n* = 4180) administrations of the LA County WIC Survey [16]. Households with more than one WIC-participating child under 5 years of age responded to the survey for one randomly selected child. Respondents received 10 United States dollars (USD) for completing a survey, and response rates exceeded 50% for each administration. Children whose mother reported more than 16 years of family WIC participation or more than 16 servings of any beverage (milk, 100% juice, water, regular soda, diet soda, sweetened <100% juice drinks, sweetened milk, and other sweetened drinks) were excluded from this study.

### 2.2. Data Sources

Survey respondents answered questions about utilization of childcare and services (health and social), health insurance, social support, household composition, food security, duration on WIC, diet, and demographics. Survey data were merged with WIC administrative data of the respondents. WIC administrative data are collected during initial program eligibility certification or annual recertification, and include directory information and characteristics of participating children (e.g., race, ethnicity, sex, and age), their mothers (e.g., education and language preference), and their households (e.g., size and income).

### 2.3. Outcome

Data on daily servings of beverages consumed by WIC-participating children between 4 and 59 months of age were obtained from parent responses to a battery of validated questions about child beverage intake [17]. Items assessed include the average daily frequency of consumption of water, sweetened milk, sweetened <100% juice drinks, other sweetened beverages (including beverages such as Gatorade or Red Bull), regular soda, and diet soda. Respondents were instructed that a serving size for sweetened beverages was “a 12-ounce can, a bottle or a glass”. Study outcomes were (A) total daily servings of any SSB, henceforth referred to as total SSB (regular soda, diet soda, sweetened <100% juice drinks, sweetened milk, and other sweetened drinks) and (B) daily servings of water. Approximately half of total daily SSB servings were sweetened <100% juice drinks, so (C) daily servings of sweetened <100% juice drinks (henceforth referred to as fruit-flavored SSBs) were evaluated as a third outcome.

### 2.4. Exposure

WIC exposure was assessed as the cumulative duration of family participation in WIC (continuous, reported in years and months). Family duration on WIC can exceed the duration of participation for the child for whom SSB and water intake is evaluated because it is inclusive of all children and pregnancies within the family. WIC offers participating families nutrition education a minimum of four times annually in addition to WIC food benefits, redeemable at approved vendors for supplemental healthy foods and beverages including fruits, vegetables, whole grains, low-fat milk, 100% juice, eggs, and cheese, which are provided monthly. Duration of WIC participation therefore served as a surrogate for each family’s cumulative WIC food package and nutrition education exposure. The self-reported duration on WIC was previously found to be valid using administrative records [15].

### 2.5. Covariates

Variables considered as potential covariates for this analysis included child age (4 to <12, 12 to <24, 24 to <36, 36 to <48, and 48 to <60 months), sex, maternally reported child race/ethnicity (Asian, Black, English-speaking Hispanic, Spanish-speaking Hispanic, and White), childcare enrollment (yes and no), and prenatal WIC enrollment (yes and no); maternal age, body mass index (BMI; weight in kg/ (height in m)^2^) based on self-reported weight and height, and education (<high school degree, high school degree, and >high school degree); household Supplemental Nutrition Assistance Program (SNAP) participation (yes and no), income (<1200, 1200 to <1800, 1800 to <2400, and ≥2400 USD per month), food security status assessed with the 6-item USDA household food security screener (high/marginal, low, and very low), number of persons (continuous), presence of an older sibling (yes and no), and presence of another WIC-participating child (yes and no); and year of survey administration (2014, 2017, and 2020). The frequency of park/playground utilization (≥3 days/week and ≤2 days/week) was included as a proxy for child physical activity.

### 2.6. Statistical Analysis

Continuous and categorical characteristics of study participants were summarized with means and standard deviations or frequencies and percentages, respectively. Differences in covariate distribution between race/ethnicity groups were assessed with chi-square tests and analysis of variance F tests for categorical and continuous covariates, respectively.

The associations between WIC exposure and child SSB and water intake, both *within* and *across* race/ethnicity groups, were assessed using multivariable negative binomial (for total SSBs and fruit-flavored SSBs, due to overdispersion in SSB outcomes) and Poisson (for water) regression models that included interaction terms between child race/ethnicity and the duration of family WIC exposure. Associations were expressed as incidence rate ratios (IRR), which were calculated *within* race/ethnicity strata comparing children from families with 2, 5 and 10 years of participation to children from families with 1 year of participation. One year was chosen as the referent value for family WIC participation because the survey design (time elapsed between construction of sampling frame and conducting surveys) precluded the inclusion of families with <6 months of WIC participation and standard WIC certification periods are of 1 year duration. IRRs were also calculated *across* race/ethnicity groups, at 1, 5 and 10 years of cumulative participation in WIC, to determine whether longer family duration on WIC reduced the magnitude of disparities in SSB and water intake *between* race/ethnicity groups. The fully adjusted negative binomial and Poisson regression models included terms for the child’s age, race/ethnicity and survey year; maternal education, age, and BMI; household size, income, food security status, and SNAP participation; family duration on WIC (linear and quadratic); and the interaction of family duration on WIC and child race/ethnicity.

Due to distinct patterns evident in associations between duration on WIC and SSB/water intake by race/ethnicity, a sensitivity analysis was undertaken to evaluate whether observed differences between Black and other race/ethnicity study participants could be attributed to SES differences between race/ethnicity groups. For these analyses, race/ethnicity was dichotomized as Black and non-Black. In negative binomial and Poisson regression models, for SSB and water intake, respectively, race was interacted with one of three dichotomous SES indicators that differed markedly between Black and non-Black study participants including household income (<1800 USD/month and ≥1800 USD/month), household SNAP participation (yes and no), and maternal educational attainment (≤high school degree and >high school degree). Dichotomous categories for household income and maternal education were determined based upon homogeneity in observed patterns of associations between family duration of WIC participation and beverage intake within categories on either side of the cut-point. The models also included interactions between race and duration on WIC, the SES indicators and duration on WIC, and three-way interactions between race, SES indicators and duration on WIC. Child activity was considered as a potential confounder, but was not related to family duration on WIC and did not meaningfully alter associations between family WIC participation duration and child beverage intake. Due to potential associations between park/playground utilization and survey year (due to COVID-19 restrictions in 2020) and the oversampling of Asian and Black participants in 2020, it was decided not to include park/playground utilization in regression models. All analyses were conducted using SAS 9.4 (SAS Institute Inc., Cary, NC, USA). *p*-values less than 0.05 were considered statistically significant.

## 3. Results

A majority of respondents to the LA County WIC survey were English- or Spanish-speaking Hispanic (46.0 and 30.9%, respectively) (Table 1). A majority of Asian survey respondents participated in the 2020 survey as anticipated due to the augmented sample in 2020, while a majority of the White respondents participated in the 2014 survey. Maternal educational attainment was higher among Asians (74.6% with >high school degree) and lower among Spanish-speaking Hispanics (43.8% with >high school degree), and household income was lower among Black respondents (52.8% <1200 USD/month). Spanish-speaking Hispanic and White respondents were the most likely to report having an older child in the household (78.6 and 72.7%, respectively). Black respondents had the highest SNAP participation (67.1%), the highest childcare utilization (48.8%), the highest mean maternal BMI (30.3 kg/m^2^), and the lowest mean household size (4.0 people).

For *within* race/ethnicity comparisons, longer family duration on WIC was associated with lower rate of daily total SSB intake among children of all races/ethnicities except for Black children (Table 2). Reductions in rate of daily total SSB intake with longer duration on WIC achieved statistical significance among English-speaking Hispanic for all durations evaluated, but protective associations were non-significant among Asian, Spanish-speaking Hispanic and White children. Higher rates of daily total SSB intake with longer family duration on WIC were non-significant among Black participants. Longer family duration on WIC was also associated with lower rates of daily fruit-flavored SSB intake among children of all races/ethnicities except among Black children. Significant reductions in the rate of daily fruit-flavored SSB intake were observed with higher reported family duration on WIC among English-speaking Hispanic children for all durations evaluated, but protective associations did not achieve statistical significance among Asian, Spanish-speaking Hispanic and White children. Longer family duration on WIC was associated with significantly higher rates of daily fruit-flavored SSB intake among Black children for all durations evaluated. Longer family duration on WIC was associated with modestly lower rates of daily water intake among all groups at all durations evaluated, and this achieved significance for English-speaking Hispanic children.

When making comparisons *across* race/ethnicity groups, disparities in rate of daily total SSB intake became non-significant or decreased in magnitude between Spanish-speaking Hispanic and English-speaking Hispanic compared to White WIC-participating children with longer family duration on WIC (Figure 1A, Appendix A); however, the disparity between Black and White WIC-participating children increased with longer family duration on WIC, and the rate of daily SSB intake was 52% higher for Black compared to White children whose families reported 10 years cumulative duration on WIC. A significant disparity in daily fruit-flavored SSB intake was observed between Black and White participants, which increased with longer family duration on WIC to a 110% higher rate of daily fruit-flavored SSB intake among Black compared to White children at 10 years of family duration on WIC (Figure 1B, Appendix A). A significant disparity was observed between rate of daily water intake between Black and White children, which persisted across family duration on WIC but did not increase in magnitude with a 14% lower rate of daily water intake among Black compared to White children at 10 years of family duration on WIC (Figure 1C, Appendix A). Spanish-speaking Hispanic children consumed water at 8% and 5% lower rates than White children at 1 and 5 years of family duration on WIC, respectively.

Similar patterns of disparities in total SSB, fruit-flavored SSB and water intake were observed in the sensitivity analyses when dichotomized race was interacted with SES indicators. For household income (Figure 2), the disparity between non-Black participants with household incomes <1800 USD/month and ≥1800 USD/month dissipated with longer WIC participation. The disparity between Black participants with household incomes <1800 USD/month and non-Black participants with household incomes ≥1800 USD/month remained consistent across the range of family WIC participation duration. The disparity between Black and non-Black participants with household incomes ≥1800 USD/month increased with longer WIC participation. A similar pattern was observed for the three beverage outcomes, and for household SNAP participation (Figure 3) and maternal educational attainment (Figure 4).

## 4. Discussion

Duration of family WIC participation is associated with healthier beverage choices among LA County WIC-participating infants and children, with lower reported intake of total and fruit-flavored SSBs with longer WIC participation duration [15]. The results of this study demonstrate that the association is not equally beneficial across racial/ethnic groups. Most groups demonstrated non-significantly lower rates of total and fruit-flavored SSB intake with longer WIC participation durations, but these were only statistically significant among English-speaking Hispanic children. It is likely that associations within groups other than English-speaking Hispanics were non-significant due to smaller sample sizes.

Persistent differences in daily SSB intake are observed nationally by race/ethnicity, with Black and Hispanic children being more likely to consume SSBs compared to White children [8]. In this study, Black children consumed total and fruit-flavored SSBs at higher rates than White children at all durations of family WIC participation, in alignment with the national data [8]. English-speaking Hispanic children consumed total SSBs at a higher rate than White children at 1 year of family duration on WIC, but the disparity dissipated with longer duration on WIC. Spanish-speaking Hispanic children consumed total and fruit-flavored SSBs at rates similar to or lower than White children at any family duration on WIC. The results of the present study, identifying no persistent disparities in intake of SSBs between Hispanic and White WIC participants, therefore, stand in contrast to the higher SSB intake observed among Hispanic children in national data [8] and suggest that WIC participation may mitigate disparities by ethnicity.

With respect to water consumption, an evaluation of 2005–2012 NHANES data reported 61% of infants 6 to 11 months and 70% of children 12 to 23 months consume water on any day [8], and the majority of children 4 to 8 years of age consume water below the recommended daily intake [9]. The average intake (4 servings per day) observed in the population of this study suggests no deficit relative to recommended intakes of 4 to 8 ounces per day for children 6 to 11 months [18], and 2 cups per day for children 1 to 3 years [19]. Children of all race/ethnicity groups in this study consumed fewer servings of water daily with longer duration of family participation in WIC, but these associations were only significant among English-speaking Hispanic children. These data suggest that family exposure to WIC services is not associated with increased water intake among participating children, in contrast to a recent educational intervention which increased water intake among children 2 to 5 years of age [20].

While racial/ethnic disparities in water intake are not evident before 24 months of age in national data [8], Black and Hispanic families are more likely than White families to report giving children bottled water because of perceived unsafe tap water [10]. The present study was unable to evaluate the type of water provided to children, but persistently lower water intake among Black children compared to White children at all durations of WIC participation suggest that racial disparities in child water intake may exist among children 4 to 59 months of age from low-income households, in contrast to prior reports of no racial/ethnic disparities in the amount of water consumed by children [8].

The increased disparity in total SSB and fruit-flavored SSB intake between children of Black and White families with increasing family duration on WIC was unexpected, as was the persistent disparity in water intake between children of Black and White families across the range of family duration on WIC. Prior research has identified significantly higher intake of SSBs among Black children in the United States relative to their White peers [21]. However, further research is needed to understand the findings of the present study, as race has not been a consistent modifier of the effects of nutrition education interventions [22] though prior perinatal educational interventions have failed to reduce the introduction of SSBs before 12 months of age among Black mothers [23].

Broader environmental conditions may contribute to increased disparities following interventions targeting health behaviors, with behavioral effects of interventions being stronger for children in less adverse neighborhood environments [24]. Prior research in the LA County WIC population identified a larger reduction in obesity prevalence in neighborhoods with higher healthy food outlet density and lower unhealthy food outlet density [25]. Differences in neighborhood environments (e.g., food environment, including food outlets and information about foods) between Black and non-Black WIC-participating children may contribute to the observed differences in the relationship between WIC participation duration and rates of SSB intake through reduced availability of healthy beverages and increased abundance of SSBs. The evaluation of disparities between Black and non-Black participants by SES identified disparities for total SSB, fruit-flavored SSB and water intake between that were of similar magnitude and persistence regardless of household SES (assessed with income, SNAP participation and maternal educational attainment). These results suggest that the identified racial disparities are not explained by different prevalence of low household SES between Black and non-Black participants.

WIC participation may influence child beverage consumption by the provision of healthy beverages in the food package, decreasing the cost of these beverages relative to others not included in WIC food packages. WIC nutrition education may improve nutritional knowledge among parents of participants, increasing perceived benefits of healthy beverages for children (e.g., water) and perceived harms of unhealthy beverages (e.g., SSBs) [26]. A study found that Hispanic parents who gave their child SSBs perceived those beverages to be healthier than parents who did not give those beverages to their child, suggesting that perceived healthfulness of beverages is important to child beverage choice among Hispanic families [27]. A recent evaluation of a countermarketing intervention for fruit-drink intake among Hispanic children found that messages about the health consequences of fruit drinks reduced intake, and joint messages about health consequences of fruit drinks and benefits of water reduced intake of fruit drinks further [28], demonstrating that joint messages about water and SSBs can enhance healthy beverage choice. Similar patterns among Hispanic families (English and Spanish speaking) in addition to Asian and White families suggest that WIC nutrition education may influence child beverage choices by providing parents with information about health consequences of excess and early introduction of dietary sugars, including SSBs. Different patterns of beverage intake observed between Black WIC-participating families and all other groups might be due in part to WIC nutrition education material not seeming culturally-relevant and not communicating effectively to Black participants. Future efforts to tailor nutrition education materials to be more culturally-relevant to diverse WIC participant populations may contribute to broader improvements in health behaviors among participants.

This study was conducted in a large and well-characterized sample of WIC-participating families in LA County, randomly-selected from the general WIC population and target geographies and racial groups, ensuring that the sample is representative of the large WIC participant population in LA County. Beverage intake outcomes were assessed with a validated instrument [17], and maternal report of family duration on WIC has been previously reported to be of acceptable validity [15]. Data on parental perception of healthfulness of beverage options were not available, limiting attempts to identify characteristics which might explain why different patterns of association were identified among race/ethnicity groups. This study was limited by the cross-sectional nature of the data, precluding assessment of the directionality of identified associations. The data are also observational, which allows for potential residual confounding by unmeasured factors and prevents causal inference. Longer WIC participation was not assigned in this observational study, and may be driven by characteristics related to both the need for nutrition assistance and receptiveness to all components of the WIC program, including the WIC food package and WIC nutrition education. If maternal characteristics related to longer participation differ between race/ethnicity groups, residual confounding by unmeasured factors related to either program need or program receptiveness may explain some of the observed pattern in disparities. The WIC service population in LA County is majority Hispanic, and results may not generalize to populations of different racial/ethnic composition.

## 5. Conclusions

Longer duration of family WIC participation may be associated with general improvement in the healthfulness of beverages selected by participating families; however, this improvement is not evident among Black families. WIC provides the same services and benefits to all participating families, and it is critical to understand whether and how WIC services may be contributing to disparities in healthy beverage choices. Further research into why Black families appear to benefit less from WIC participation should be prioritized. Child beverage choice may need to be given greater emphasis during individual nutrition counseling efforts for Black families. Further evaluation of WIC services for differential impacts across race/ethnicity groups may also need to be considered.

## Figures and Tables

**Figure 1 nutrients-14-01048-f001:**
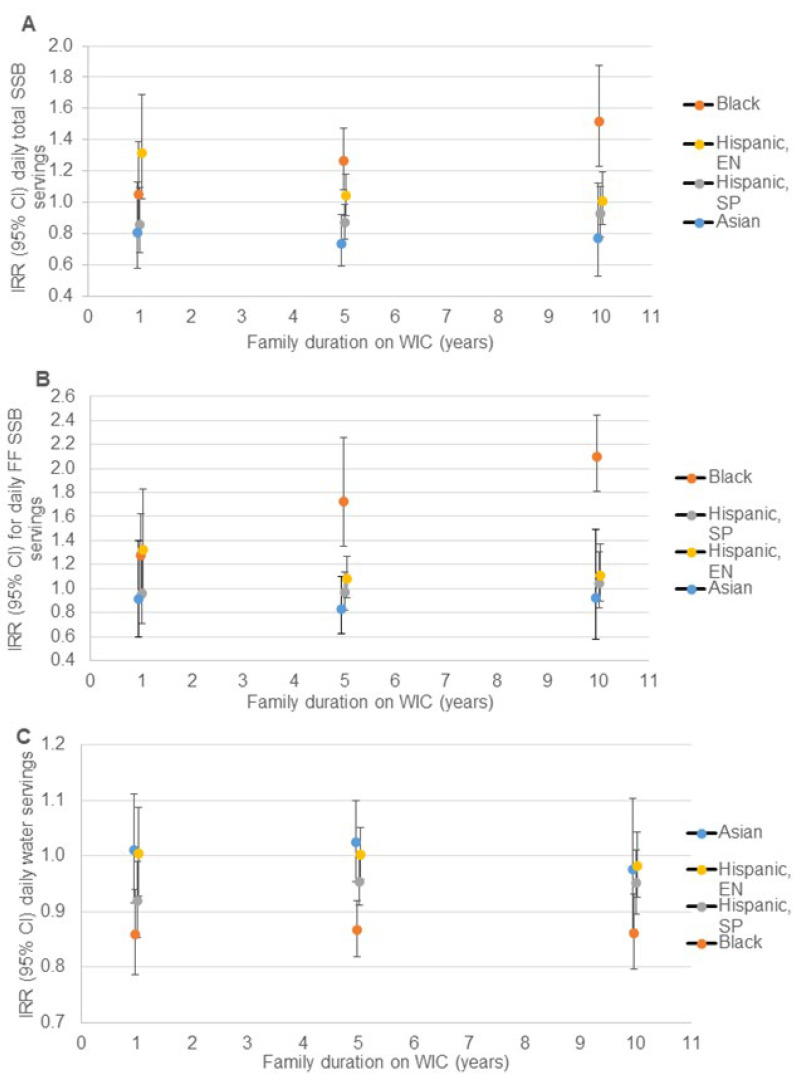
Relative rate of daily beverage intake for non-White compared to White children at 1, 5 and 10 years of family WIC participation (*n* = 11,482): (**A**) total SSB ^a^; (**B**) fruit-flavored SSB ^a^; (**C**) water ^b^. CI = confidence interval; EN = English speaking; IRR = incidence rate ratio; SP = Spanish speaking; SSB = sugar-sweetened beverage; WIC = the Special Supplemental Nutrition program for Women, Infants and Children. ^a^ Estimates are IRR (95% CI) for daily servings of each beverage comparing non-White to White children at 1, 5 and 10 years of WIC participation from negative binomial regression models including terms for the child’s age, race/ethnicity and survey year; maternal education, age, and BMI; household size, income, food security status, SNAP participation; family duration on WIC (linear and quadratic); and the interaction of family duration on WIC and child race/ethnicity. CI is represented by vertical bars for each point estimate. ^b^ Estimates are IRR (95% CI) for daily servings of each beverage comparing non-White to White children at 1, 5 and 10 years of WIC participation from Poisson regression models including terms for the child’s age, race/ethnicity and survey year; maternal education, age, and BMI; household size, income, food security status, SNAP participation; family duration on WIC (linear and quadratic); and the interaction of family duration on WIC and child race/ethnicity. CI is represented by vertical bars for each point estimate.

**Figure 2 nutrients-14-01048-f002:**
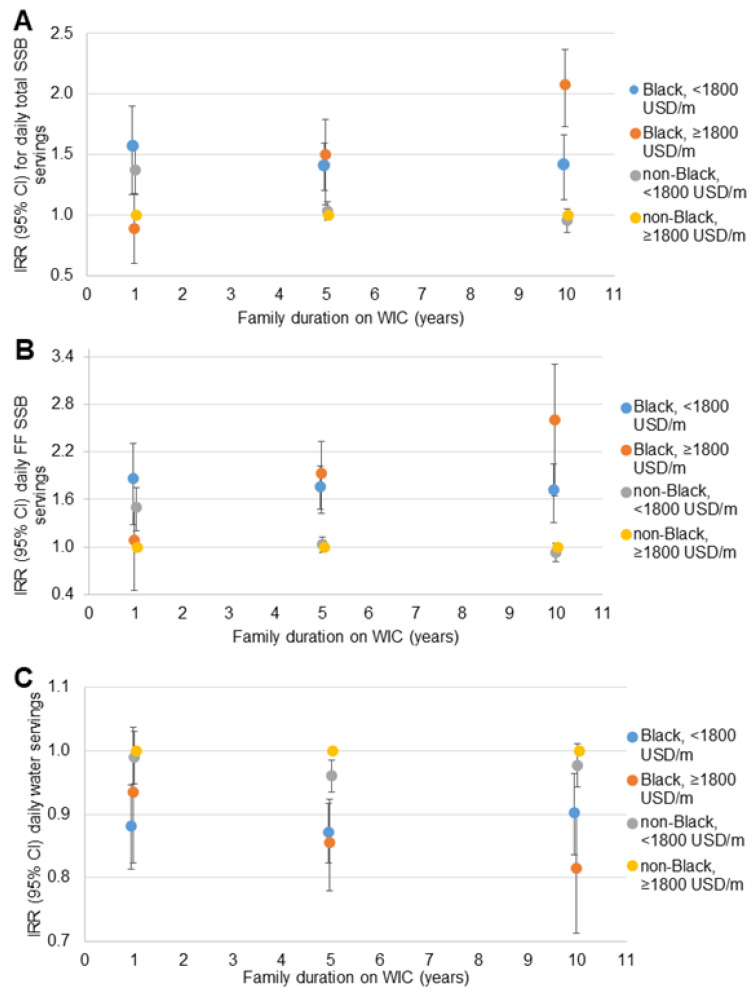
Relative rate of daily beverage intake compared to non-Black children with household income ≥1800 USD/month at 1, 5 and 10 years of family WIC participation (*n* = 11,482): (**A**) total SSB ^a^; (**B**) fruit-flavored SSB ^a^; (**C**) water ^b^. CI = confidence interval; IRR = incidence rate ratio; m = month; SSB = sugar-sweetened beverage; USD= United States Dollars; WIC = the Special Supplemental Nutrition program for Women, Infants and Children. ^a^ Estimates are IRR (95% CI) for daily servings of each beverage compared to non-Black children with household income ≥1800 USD/month at 1, 5 and 10 years of WIC participation from negative binomial regression models including terms for the child’s age, race/ethnicity and survey year; maternal education, age, and BMI; household size, income (dichotomous), food security status, SNAP participation; family duration on WIC (linear and quadratic); the interaction of family duration on WIC and child race/ethnicity; the interaction of dichotomous household income and family duration on WIC; and the three-way interaction of household income, child race/ethnicity and family duration on WIC. CI is represented by vertical bars for each point estimate. ^b^ Estimates are IRR (95% CI) for daily servings of each beverage compared to non-Black children with household income ≥1,800 USD/month at 1, 5 and 10 years of WIC participation from Poisson regression models including terms for the child’s age, race/ethnicity and survey year; maternal education, age, and BMI; household size, income (dichotomous), food security status, SNAP participation; family duration on WIC (linear and quadratic); the interaction of family duration on WIC and child race/ethnicity; the interaction of dichotomous household income and family duration on WIC; and the three-way interaction of household income, child race/ethnicity and family duration on WIC. CI is represented by vertical bars for each point estimate.

**Figure 3 nutrients-14-01048-f003:**
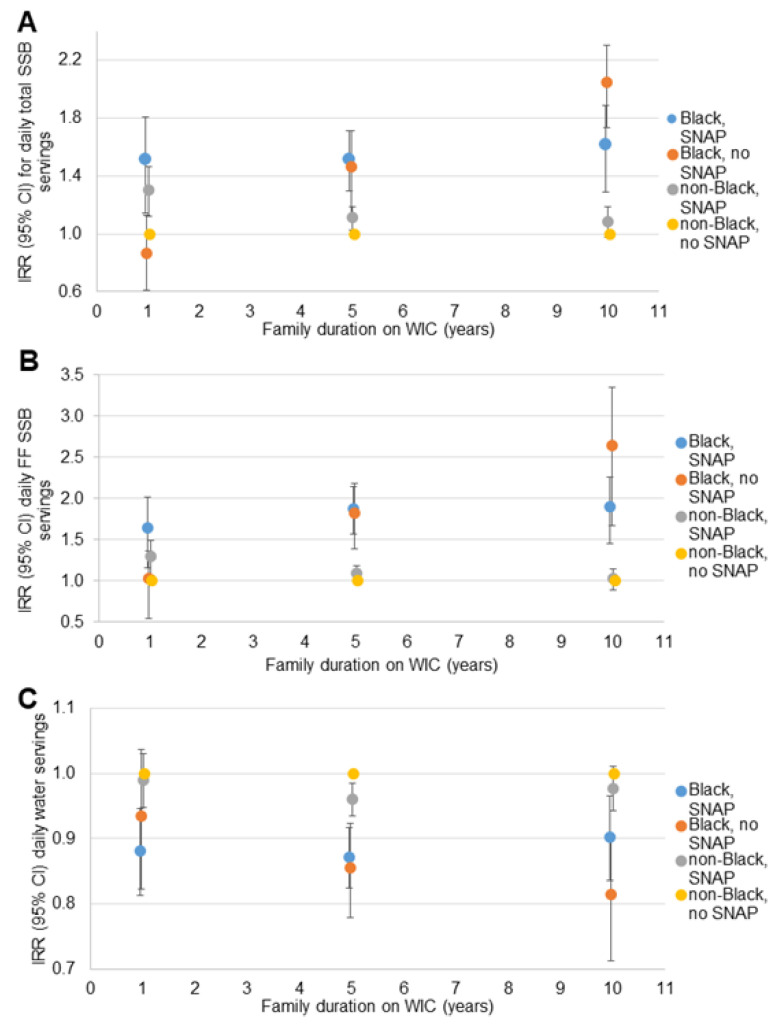
Relative rate of daily beverage intake compared to non-Black children with no household SNAP participation at 1, 5 and 10 years of family WIC participation (*n* = 11,482): (**A**) total SSB ^a^; (**B**) fruit-flavored SSB ^a^; (**C**) water ^b^. CI = confidence interval; IRR = incidence rate ratio; SNAP = Supplemental Nutrition Assistance Program; SSB = sugar-sweetened beverage; WIC = the Special Supplemental Nutrition program for Women, Infants and Children. ^a^ Estimates are IRR (95% CI) for daily servings of each beverage compared to non-Black children who do not participate in SNAP children at 1, 5 and 10 years of WIC participation from negative binomial regression models including terms for the child’s age, race/ethnicity and survey year; maternal education, age, and BMI; household size, income, food security status, SNAP participation; family duration on WIC (linear and quadratic); the interaction of family duration on WIC and child race/ethnicity; the interaction of SNAP participation and family duration on WIC; and the three-way interaction of SNAP participation, child race/ethnicity and family duration on WIC. CI is represented by vertical bars for each point estimate. ^b^ Estimates are IRR (95% CI) for daily servings of each beverage compared to non-Black children who do not participate in SNAP at 1, 5 and 10 years of WIC participation from Poisson regression models including terms for the child’s age, race/ethnicity and survey year; maternal education, age, and BMI; household size, income, food security status, SNAP participation; family duration on WIC (linear and quadratic); the interaction of family duration on WIC and child race/ethnicity; the interaction of SNAP participation and family duration on WIC; and the three-way interaction of SNAP participation, child race/ethnicity and family duration on WIC. CI is represented by vertical bars for each point estimate.

**Figure 4 nutrients-14-01048-f004:**
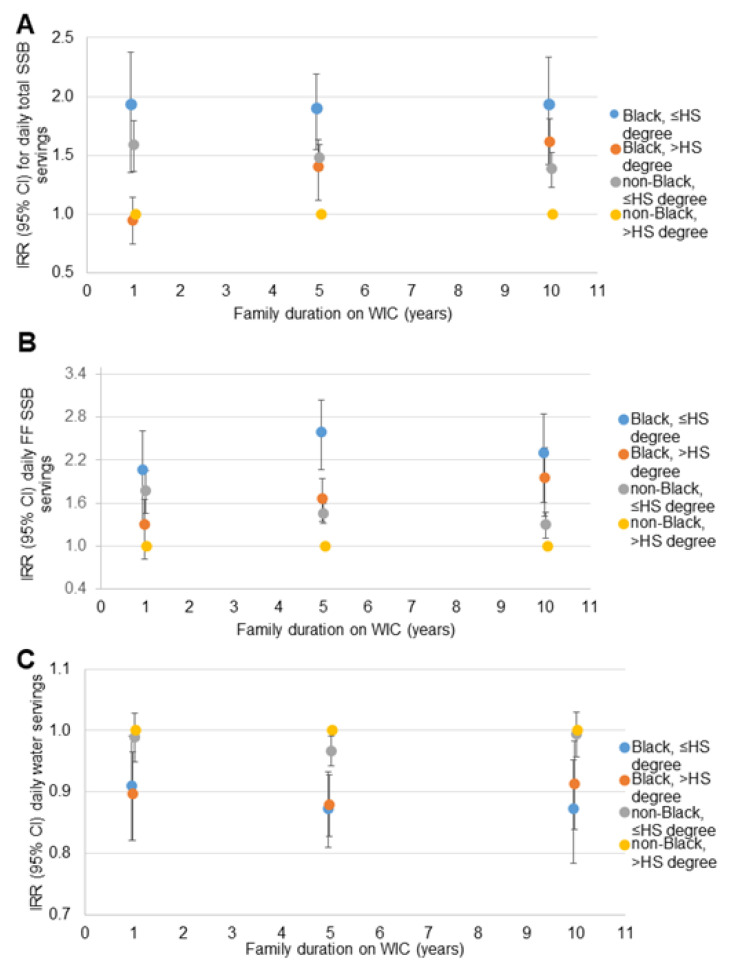
Relative rate of daily beverage intake compared to non-Black children with a mother with >HS degree at 1, 5 and 10 years of family WIC participation (*n* = 11,482): (**A**) total SSB ^a^; (**B**) fruit-flavored SSB ^a^; (**C**) water ^b^. CI = confidence interval; HS = high school; IRR = incidence rate ratio; SSB = sugar-sweetened beverage; WIC = the Special Supplemental Nutrition program for Women, Infants and Children. ^a^ Estimates are IRR (95% CI) for daily servings of each beverage compared to non-Black children with higher maternal education at 1, 5 and 10 years of WIC participation from negative binomial regression models including terms for the child’s age, race/ethnicity and survey year; maternal education (dichotomous), age, and BMI; household size, income, food security status, SNAP participation; family duration on WIC (linear and quadratic); the interaction of family duration on WIC and child race/ethnicity; the interaction of maternal education and family duration on WIC; and the three-way interaction of maternal education, child race/ethnicity and family duration on WIC. CI is represented by vertical bars for each point estimate. ^b^ Estimates are IRR (95% CI) for daily servings of each beverage compared to non-Black children with higher maternal education at 1, 5 and 10 years of WIC participation from Poisson regression models including terms for the child’s age, race/ethnicity and survey year; maternal education (dichotomous), age, and BMI; household size, income, food security status, SNAP participation; family duration on WIC (linear and quadratic); the interaction of family duration on WIC and child race/ethnicity; the interaction of maternal education and family duration on WIC; and the three-way interaction of maternal education, child race/ethnicity and family duration on WIC. CI is represented by vertical bars for each point estimate.

**Table 1 nutrients-14-01048-t001:** Child, maternal and household factors by WIC participation duration for children aged 4 to 59 months from 2014, 2017 and 2020 LA County WIC Surveys (*n* = 11,482) ^a^.

Variable		Asian	Black	Hispanic, EN	Hispanic, SP	White	
		*N* = 634	*N* = 1141	*N* = 5281	*N* = 3548	*N* = 878	*p*-Value
**Child characteristics**							
Male		337 (53.2)	545 (47.8)	2699 (51.1)	1840 (51.9)	447 (50.9)	0.14
Age							<0.0001
	4 to <12 m	83 (13.1)	171 (15.0)	770 (14.6)	391 (11.0)	99 (11.3)	
	12 to <24 m	184 (29.0)	297 (26.0)	1393 (26.4)	718 (20.2)	197 (22.4)	
	20 to <36 m	142 (22.4)	240 (21.0)	1145 (21.7)	769 (21.7)	184 (21.0)	
	36 to <48 m	133 (21.0)	239 (20.9)	1073 (20.3)	858 (24.2)	203 (23.1)	
	48 to <60 m	92 (14.5)	194 (17.0)	900 (17.0)	812 (22.9)	195 (22.2)	
Prenatal enrollment		255 (40.2)	401 (35.1)	1568 (29.7)	1073 (30.2)	314 (35.8)	<0.0001
Current childcare use		155 (24.6)	556 (48.8)	2349 (44.5)	883 (24.9)	257 (29.3)	<0.0001
Daily servings							
Water		4.1 ± 2.3	3.5 ± 2.0	3.9 ± 2.2	4.1 ± 2.1	4.1 ± 2.2	<0.0001
Total SSB		0.5 ± 0.9	1.0 ± 1.5	0.7 ± 1.2	1.0 ± 1.4	0.9 ± 1.3	<0.0001
Fruit-flavored SSB		0.2 ± 0.6	0.5 ± 0.9	0.3 ± 0.7	0.4 ± 0.7	0.4 ± 0.7	<0.0001
Park/playground visits							<0.0001
	≥3 days/week	250 (39.5)	638 (56.2)	3340 (63.4)	1823 (51.5)	447 (50.9)	
	≤2 days/week	383 (60.5)	497 (43.8)	1931 (36.6)	1719 (48.5)	431 (49.1)	
Survey year							<0.0001
	2014	26 (4.1)	377 (33.0)	1769 (33.5)	952 (26.8)	520 (59.2)	
	2017	78 (12.3)	246 (21.6)	1915 (36.3)	1308 (36.9)	111 (12.6)	
	2020	530 (83.6)	518 (45.4)	1597 (30.2)	1288 (36.3)	247 (28.1)	
**Maternal characteristcs**							
Maternal education							<0.0001
	<HS	23 (3.6)	113 (9.9)	769 (14.6)	1994 (56.2)	294 (33.5)	
	HS grad	138 (21.8)	359 (31.5)	1673 (31.7)	999 (28.2)	246 (28.0)	
	>HS	473 (74.6)	669 (58.6)	2839 (53.8)	555 (15.6)	338 (38.5)	
Maternal BMI		26.4 ± 6.0	30.3 ± 7.6	29.4 ± 6.7	28.7 ± 6.2	28.2 ± 6.4	<0.0001
**Household characteristics**							
Food security							<0.0001
	Low	138 (21.8)	268 (23.5)	956 (18.1)	821 (23.1)	203 (23.1)	
	Very low	22 (3.5)	100 (8.8)	310 (5.9)	203 (5.7)	76 (8.7)	
Income, USD/m							<0.0001
	<1200	148 (23.3)	602 (52.8)	1939 (36.7)	899 (25.3)	309 (35.2)	
	1200 to <1800	167 (26.3)	245 (21.5)	1390 (26.3)	1415 (39.9)	271 (30.9)	
	1800 to <2400	174 (27.4)	176 (15.4)	1025 (19.4)	738 (20.8)	171 (19.5)	
	≥2400	145 (22.9)	118 (10.3)	927 (17.6)	496 (14.0)	127 (14.5)	
Older child present		367 (57.9)	719 (63.0)	3470 (65.7)	2790 (78.6)	638 (72.7)	<0.0001
Multiple WIC children		140 (22.1)	278 (24.4)	1291 (24.4)	702 (19.8)	189 (21.5)	<0.0001
SNAP		250 (39.4)	766 (67.1)	2247 (42.5)	1674 (47.2)	462 (52.6)	<0.0001
Household size		4.5 ± 1.5	4.0 ± 1.5	4.6 ± 1.6	4.8 ± 1.5	4.6 ± 1.5	<0.0001
Duration on WIC, m		3.5 ± 2.6	4.7 ± 3.7	4.6 ± 3.4	6.3 ± 3.9	5.6 ± 3.7	<0.0001

Footnote: BMI = body mass index; EN = English speaking; HS = High school; LA = Los Angeles; M = months; SNAP = Supplemental Nutrition Assistance Program; SP = Spanish speaking; USD = United States Dollar; SSB = sugar-sweetened beverage; WIC = Special Supplemental Nutrition Program for Women, Infants and Children; ^a^
*p*-values for comparisons between race/ethnicity groups were calculated with Chi-square or analysis of variance F-tests, for categorical and continuous variables, respectively.

**Table 2 nutrients-14-01048-t002:** Relative rate of daily intake of total SSBs, fruit-flavored SSBs and water for children with 2, 5 and 10 years compared to 1 years of family WIC participation by race/ethnicity (*n* = 11,482).

		1 Year	2 Years ^c^	5 Years ^d^	10 Years ^e^
Total SSB ^a^					
	Asian	1.00	0.96 (0.85, 1.08)	0.87 (0.62, 1.23)	0.83 (0.52, 1.33)
	Black	1.00	1.04 (0.97, 1.11)	1.14 (0.93, 1.40)	1.26 (0.96, 1.64)
	Hispanic, SP	1.00	0.99 (0.95, 1.02)	0.96 (0.86, 1.08)	0.94 (0.81, 1.09)
	Hispanic, EN	1.00	0.92 (0.88, 0.95)	0.75 (0.67, 0.85)	0.67 (0.57, 0.79)
	White	1.00	0.99 (0.92, 1.07)	0.95 (0.75, 1.21)	0.87 (0.63, 1.20)
Fruit-flavored SSB ^a^					
	Asian	1.00	0.96 (0.83, 1.12)	0.89 (0.57, 1.39)	0.89 (0.49, 1.62)
	Black	1.00	1.09 (1.02, 1.18)	1.33 (1.06, 1.68)	1.45 (1.07, 1.96)
	Hispanic, SP	1.00	1.00 (0.95, 1.04)	0.99 (0.86, 1.14)	0.96 (0.80, 1.15)
	Hispanic, EN	1.00	0.94 (0.89, 0.98)	0.80 (0.69, 0.94)	0.73 (0.60, 0.90)
	White	1.00	1.00 (0.91, 1.11)	0.98 (0.72, 1.34)	0.88 (0.59, 1.32)
Water ^b^					
	Asian	1.00	0.99 (0.96, 1.02)	0.97 (0.88, 1.06)	0.93 (0.81, 1.06)
	Black	1.00	0.99 (0.96, 1.01)	0.96 (0.90, 1.04)	0.96 (0.88, 1.06)
	Hispanic, SP	1.00	1.00 (0.98, 1.01)	0.99 (0.95, 1.02)	0.99 (0.95, 1.04)
	Hispanic, EN	1.00	0.98 (0.97, 1.00)	0.95 (0.91, 0.99)	0.94 (0.89, 0.99)
	White	1.00	0.98 (0.96, 1.01)	0.95 (0.88, 1.03)	0.96 (0.86, 1.06)

Footnote: CI = confidence interval; EN = English speaking; IRR = incidence rate ratio; SP = Spanish speaking; SSB = sugar-sweetened beverage; WIC = the Special Supplemental Nutrition program for Women, Infants and Children. ^a^ Estimates are from negative binomial regression models including terms for the child’s age, race/ethnicity and survey year; maternal education, age, and BMI; household size, income, food security status, SNAP participation; family duration on WIC (linear and quadratic); and the interaction of family duration on WIC and child race/ethnicity. ^b^ Estimates are from Poisson regression models including terms for the child’s age, race/ethnicity and survey year; maternal education, age, and BMI; household size, income, food security status, SNAP participation; family duration on WIC (linear and quadratic); and the interaction of family duration on WIC and child race/ethnicity. ^c^ IRR (95% CI) for daily servings of each beverage comparing children of families with 2 years of WIC participation to those with 1 year of WIC participation. ^d^ IRR (95% CI) for daily servings of each beverage comparing children of families with 5 years of WIC participation to those with 1 year of WIC participation. ^e^ IRR (95% CI) for daily servings of each beverage comparing children of families with 10 years of WIC participation to those with 1 year of WIC participation.

## Data Availability

Confidential data were used in this study, and will not be made available due to an agreement with the California Department of Public Health WIC Division.

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
