# Peer review of "Longer Participation in the Special Supplemental Nutrition Program for Women, Infants, and Children Is Not Associated with Reduced Sugar-Sweetened Beverage Intake among Black Participants"

_nutrients, 2022, doi:10.3390/nu14051048_

Round 1

Reviewer 1 Report

The manuscript "Longer duration on WIC not associated with reduced sugar-2 sweetened beverage intake among Black participants" contains the results of an important study which was carried out on a large number of participants. The isse treated is also of great importance.

The manuscript, however, has some shortcomings and its reading is not fluent.

WIC in title has to be substituted with the extensive name of the project “Special Supplemental Nutrition Program for Women, Infants and Children (WIC)” or you may change it in something like “The efficacy of supplementation with … in reducing … by ethnicity etc.”

Overall: use the form “we assessed, we analyzed…” instead of “was assessed, was analyzed…”

The manuscript is very long and it is not very comprehensive on firs reading.

I would suggest as it follows:

  • Information given in lines 61-67 should be given at the very beginning. “Data have suggested that WIC-participant SSB consumption has decreased (14), and a recent study identified significant reductions in SSB intake among WIC-participating children of families with longer (specify how long/much longer!) duration on the program (15). The present study was conducted to assess whether SSB intake is reduced and whether water intake is increased with longer family duration on WIC within all racial/ethnic groups, and whether disparities in SSB and water intake 66 between race/ethnicity groups would be reduced by longer family duration on WIC) OR how the duration of participation in the program affects SSB and water intake within and between different racial and ethnic groups”.
  • You should quantify the problem: how much SSB are consumed by children of different age normally (overall and for different groups), what kind of SSB, what does that mean, do you have any national data on correlation between SSB consumption and diabetes?
  • You should explain well in what the WIC program consists: who is enrolled, based on which variables (income and/or other conditions), how (national agencies call based on…? or subjects must ask to participate themselves…), what kind of food is provided (kcals, proteins, just a snack or what, every day or when…), how it is provided…

Abstract:

Line 16 – relationship no relationships. Isn’t effect better than relationship?

Line 21 - Dose of WIC service received? I don’t understand (do you mean dose of supplements or duration of program?)

“…with 10 years” Do you mean “that have participated for 10 years”?  Sorry if I do not understand but there will be many nonnative English speakers

English-speaking Hispanic children of families with 10 years on WIC consumed 33 and 27%????  fewer servings of total and fruit-flavored SSBs compared to those of 24 families with 1 year on WIC. – Too messy

Black children from families with 5 and 10 years of participation in 25 WIC consumed 33 and 45% more daily servings of fruit-flavored SSB than those (black or all?) from families with 26 1 year on WIC

Introduction

Lines 46-55, could you add some numbers? How important are disparities, earnings…?

Line 111- explain what they receive

Results

Put your results in table or in text, do not repeat the results which can be red in table in the text, just comment shortly. That will help make your text shorter and easier to read and understand.

Discussion

Very long sentences, untypical for English and difficult to follow the sense!

Information on water intake should go at the beginning of the manuscript together with similar information on SSB.

Can you suggest what part of WIC program influences the improvement in the healthfulness of beverages selected by participating families: education or food provided that substitutes unhealthy SSB?

I think that the main problem is not the ethnicity but the main socio-economic and cultural characteristics, which characterize participants of different ethnicities. I suppose that children who participate are all from low income families (but maybe it is not so, you do not explain it and you should). So if your results should be used to intervene and solve the problem, it is for sure that the point is not the color of the skin but issues like familiarity (other family members overweight or obese), different eating habits and traditional dishes, their preferences, education, income which is probable lower than average…

You should end the manuscript with indications on:

  1. Limits of your study
  2. Suggestions for future studies
  3. Indications on transferability of your results

Reviewer 2 Report

The submitted manuscript aims at assessing whether sugar-sweetened beverages (SSB) intake is reduced and whether water intake is increased with longer family duration on WIC within all racial/ethnic groups. Also whether disparities in SSB and water intake between race/ethnicity groups would be reduced by longer family duration on WIC. This cross-sectional study using WIC participants from the largest local population of WIC participants in the US. The results suggest that longer duration of family WIC participation may be associated with general improvement in the healthfulness of beverages selected by participating families; however, this improvement is not evident among Black families. The aim of the study is relevant to Nutrients’ special issue “ The Influence of Social Determinants, Nutrition Policy on
Healthy Eating Lifestyle”. The hypotheses of this manuscript were well tested and answered. A few limitations which limit the relevance and the interest of the results.

  1. It is not clear how to categorize children into different race/ethnicity? For example, if parents are from different race/ethnicity.
  2. Is there any exclusion criteria in this study? For example, If one or more children in a household will they all include in the study?
  3. Please define “Total SSB”.
  4. Milk intake still contribute to a significant amount of total fluid consumption in this age range of children. This manuscript didn’t provide any information about milk consumption among these children.
  5. Children between 4 and 59 months of age, the activity level is a major confounder when assess energy or water intake. Is there any information provide in this study?
  6. Table 1, please provide the statistic method used in the footnote.
  7. Table 2, for within race/ethnicity comparisons, I think for both negative binomial regression and Poisson regression models should not include “race/ethnicity” in the model (Table 2 footnote)

Round 2

Reviewer 1 Report

I appreciate your work which is now ready to be published!